# META KNOWLEDGE CONDENSATION FOR FEDERATED LEARNING

**Ping Liu** *
Center for Frontier AI Research
A*STAR
138632, Singapore
{pino.pingliu}@gmail.com

**Xin Yu** *
Information Technology and Electrical Engineering
The University of Queensland
Brisbane, Australia
{xin.yu}@uq.edu.au

**Joey Tianyi Zhou**[†]
Center for Frontier AI Research
A*STAR
138632, Singapore
{joey_zhou}@cfar.a-star.edu.sg

## ABSTRACT

Existing federated learning paradigms usually extensively exchange distributed models at a central solver to achieve a more powerful model. However, this would incur severe communication burden between a server and multiple clients especially when data distributions are heterogeneous. As a result, current federated learning methods often require a large number of communication rounds in training. Unlike existing paradigms, we introduce an alternative perspective to significantly decrease the communication cost in federate learning. In this work, we first introduce a meta knowledge representation method that extracts meta knowledge from distributed clients. The extracted meta knowledge encodes essential information that can be used to improve the current model. As the training progresses, the contributions of training samples to a federated model also vary. Thus, we introduce a dynamic weight assignment mechanism that enables samples to contribute adaptively to the current model update. Then, informative meta knowledge from all active clients is sent to the server for model update. Training a model on the combined meta knowledge without exposing original data among different clients can significantly mitigate the heterogeneity issues. Moreover, to further ameliorate data heterogeneity, we also exchange meta knowledge among clients as conditional initialization for local meta knowledge extraction. Extensive experiments demonstrate the effectiveness and efficiency of our proposed method. Remarkably, our method outperforms the state-of-the-art by a large margin (from 74.07% to 92.95%) on MNIST with a restricted communication budget (*i.e.*, 10 rounds).

## 1 INTRODUCTION

Most deep learning-based models are trained in a data-centralized manner. However, in some cases, data might be distributed among different clients and cannot be shared. To address this issue, Federated Learning (FL) (Yang et al., 2019b;a; Kairouz et al., 2021) has been proposed to learn a powerful model without sharing private original data among clients. In general, most prior FL works often require frequent model communications to exchange models between local clients and a global server, resulting in heavy communications burden (Wu & Wang, 2021; Chencheng et al., 2022). Therefore, it is highly desirable to obtain a powerful federated model with only a few communication rounds.

In this work, we propose a new meta knowledge-driven federated learning approach to achieve an effective yet communication-efficient model, thus significantly reducing communication costs. Unlike

---

*Equal contributions
[†]corresponding author

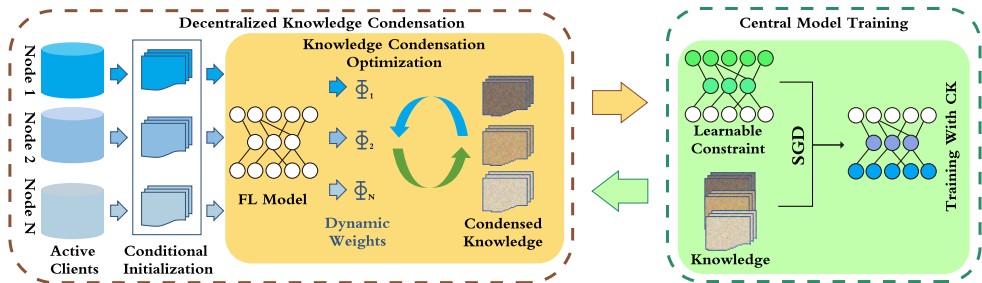

**Figure 1:** Illustration of our pipeline, in which only three active clients are shown. The local clients conduct meta knowledge condensation from local private data, and the server utilizes the uploaded meta knowledge for training a global model. The local meta knowledge condensation and central model training are conducted in an iterative manner. For meta knowledge extraction on clients, we design two mechanisms, *i.e.*, meta knowledge sharing, and dynamic weight assignment. For server-side central model training, we introduce a learnable constraint.

prior works, we formulate federated learning in a new perspective, where representative information will be distilled from original data and sent to the server for model training. On the client side, we extract representative information of original data and condense it into a tiny set of highly-compressed synthetic data, namely *meta knowledge*. Furthermore, we develop two mechanisms, *i.e.*, dynamic weight assignment and meta knowledge sharing, in the condensation process to mitigate the data heterogeneity issue widely existing in decentralized data. On the server side, we train our global model with meta knowledge uploaded from clients rather than simply averaging client models.

Specifically, we firstly distill the task-specific knowledge from private data on local clients and condense it as meta knowledge. The meta knowledge condensation process is modeled as a bi-level optimization procedure under the federated learning setting: the inner-loop minimizes the training loss on meta knowledge to update a model; and the outer-loop minimizes the training loss on original data to update meta knowledge based on the updated model. In the optimization process, we assign dynamic weights to each sample based on its training loss. By dynamically adjusting the weight of each sample in training, we empower each sample to contribute adaptively to the current model. Besides, to further mitigate heterogeneous data distributions among different clients, we design a meta knowledge sharing mechanism.

Our model can be trained with meta knowledge of various clients, which better describes the overall distribution. This is in contrast to previous methods that average local models on the server. To further improve the stability of the central model training, we incorporate a learnable conditional generator. The generator models the statistical distribution of the uploaded meta knowledge and generates synthetic samples, which provide historical information to the model update. It is worth noting that meta knowledge, which contains the essential information of the original data and the corresponding class information, can be used as normal training data for model training. As a result, our global model is trained with both the uploaded and generated meta knowledge on the server side, effectively reducing the impact of data heterogeneity and reducing the number of communication rounds.

We have conducted extensive experiments on several benchmark datasets, including MNIST (LeCun et al., 2010), SVHN (Netzer et al., 2011), CIFAR10 (Krizhevsky & Hinton, 2009), and CIFAR100 (Krizhevsky & Hinton, 2009). The results demonstrate the efficacy and efficiency of our proposed approach. In particular, our method demonstrates a significant improvement over the competing works, particularly in scenarios with limited communication budgets (*i.e.*, 10 communication rounds). Overall, our key contributions are summarized as follows:

- We propose a new meta knowledge driven federated learning approach, in which we present a novel approach for federated meta-knowledge extraction. Our method can effectively encodes local data for global model training. Specifically, we formulate a dynamic weight assignment mechanism to enhance the informative content of the extracted meta-knowledge, and design a knowledge sharing strategy to facilitate the exchange of meta-knowledge among clients without exchanging the original data.

- We introduce a server-side conditional generator that models the statistical distribution of uploaded meta knowledge to stabilize the training process. Benefiting from the extracted

meta knowledge and learned statistical distribution, our model requires fewer communication rounds compared to competing methods while achieving superior performance.

## 2 METHODOLOGY

### 2.1 PROBLEM DEFINITION

Federated Learning (FL) trains a model across a set of decentralized devices, *i.e.*, a set of clients and a server. Suppose there is data $\mathcal{D}$ distributed on $C$ clients, each of which has a local private training dataset $\mathcal{D}^c = \{x_i^c, y_i^c\}$, $1 \le i \le n^c$ and a weight value $p^c$. It is noted that $\mathcal{D} = \cup \mathcal{D}^c$ and $\sum_{c=1}^{C} p^c = 1$.

Without loss of generality, we discuss a multi-class classification problem under a federated learning setting. The learning target is formulated as follows:

$$\min_{\mathbf{w}} \{\mathcal{L}(\mathbf{w}, \mathcal{D}) \triangleq \sum_{c=1}^{C} p^c \mathcal{L}^c(\mathbf{w}, \mathcal{D}^c)\}, \tag{1}$$

where $\mathcal{L}^c(\cdot)$ is a local objective function that is optimized on the $c$-$th$ client. The loss function $\mathcal{L}^c(\cdot)$ is formulated as follows:

$$\mathcal{L}^c(\mathbf{w}, \mathcal{D}^c) \triangleq \frac{1}{n^c} \sum_{i=1}^{n^c} \ell(\mathbf{w}, x_i^c, y_i^c), \tag{2}$$

where $\ell(\cdot)$ is a user-defined loss function (*e.g.*, cross-entropy loss), and $\mathbf{w}$ denotes model parameters. As most FL algorithms need to exchange locally trained models multiple times, communication burden is often non-negligible. Though one-shot FL (Zhou et al., 2020a) has been proposed to reduce communication cost, it suffers performance degradation.

### 2.2 FEDERATED LEARNING VIA META KNOWLEDGE

We propose a new FL approach to solve the aforementioned limitations. Specially, as shown in Figure 1, our method conducts federated meta knowledge extraction (FMKE) on local clients and server-side central model training (CMT). To mitigate the data heterogeneity issue in FMKE, we design two mechanisms, *i.e.*, dynamic weight assignment, and meta knowledge sharing. To stabilize the central model training, we introduce a learnable constraint modeled by a conditional generator. The technical details are provided in the following subsections.

#### 2.2.1 FEDERATED META KNOWLEDGE EXTRACTION ON CLIENTS

We design Federated Meta Knowledge Extraction (FMKE) to extract key information from decentralized data. In the decentralized scenario, the original data $\mathcal{D}$ is distributed on a set of clients. Each client has its private data $\mathcal{D}^c$ ($\mathcal{D} = \cup \mathcal{D}^c$, $1 \le c \le C$, where $C$ is the client number), and a model downloaded from a server. For simplifying the following discussion, we denote the model downloaded by client $c$ as $\mathbf{w}^c$.

On each active local client $c$, FKME distills key information from corresponding local private data $\mathcal{D}^c$. The distilled information is condensed as meta knowledge $\hat{\mathcal{D}}^c$, which will be used to replace the original data $\mathcal{D}^c$ in global model training *on the server* [1]. The cardinality of $\hat{\mathcal{D}}$, *i.e.*, the size of extracted meta knowledge from all active clients, is much less than that of $\mathcal{D}$. The condensed meta knowledge $\hat{\mathcal{D}}$ is highly compressed and representative. To enable the models trained on the original data and the meta knowledge to achieve the comparable performance, we formulate the objective function of FMKE as[2]:

$$\hat{\mathcal{D}}^{c,*} = \arg \min_{\hat{\mathcal{D}}^c} \mathcal{L}^c(\mathbf{w}^*, \mathcal{D}^c) = \arg \min_{\hat{\mathcal{D}}^c} \mathcal{L}^c(\mathbf{w}^c - \eta \nabla_{\mathbf{w}^c} \mathcal{L}^c(\mathbf{w}^c, \hat{\mathcal{D}}^c), \mathcal{D}^c), \tag{3}$$

where $\eta$ denotes a learning rate for updating $\mathbf{w}^c$ by stochastic gradient descent (SGD).

To solve the above objective function, we employ a bi-level optimization solution (Rajeswaran et al., 2019; Wang et al., 2018). To be specific, the meta knowledge extraction is formulated as a nested optimization process: in the inner-loop, based on an initialized meta knowledge, a model is updated to minimize the training loss over the meta knowledge; in the outer-loop, given the updated model,

---

[1]The dimension of synthesized $\hat{\mathcal{D}}_i^c$ is the same as that of original data $\mathcal{D}_i^c$.

[2]Assume one gradient descent step is conducted to achieve $\mathbf{w}^*$ here.

the meta knowledge is renewed by minimizing the training loss over the original data. The iterative formulation of our bi-level optimization process on client $c$ is written as:

$$\hat{\mathcal{D}}^{c,*} = \arg\min_{\hat{\mathcal{D}}^c} \mathcal{L}^c(\mathbf{w}^*, \mathcal{D}^c) \quad s.t. \quad \mathbf{w}^* = \arg\min_{\mathbf{w}^c} \mathcal{L}^c(\mathbf{w}^c, \hat{\mathcal{D}}^c), \quad (4)$$

where $\mathcal{L}^c(\cdot, \mathcal{D}^c)$ denotes a loss function over the original data $\mathcal{D}^c$ on client $c$, $\mathcal{L}^c(\cdot, \hat{\mathcal{D}}^c)$ denotes a loss function over the meta knowledge $\hat{\mathcal{D}}^c$ on client $c$.

The inner-loop and outer-loop are implemented alternatingly and stochastic gradient descent is employed to update the model and meta knowledge. At first, on an active client $c$, we update the model parameter by the following formulation:

$$\mathbf{w}^c \leftarrow \mathbf{w}^c - \eta \nabla_{\mathbf{w}^c} \mathcal{L}^c(\mathbf{w}^c, \hat{\mathcal{D}}^c), \quad (5)$$

where $\eta$ denotes a learning rate for the inner-loop. An updated model $\mathbf{w}^*$ is obtained in this inner-loop.

Secondly, given the updated model $\mathbf{w}^*$, we evaluate it on original data $\mathcal{D}^c$ and calculate the loss $\mathcal{L}^c(\mathbf{w}^*, \mathcal{D}^c)$. Then the condensed meta knowledge $\hat{\mathcal{D}}^c$ can be updated by:

$$\hat{\mathcal{D}}^c \leftarrow \hat{\mathcal{D}}^c - \alpha \nabla_{\hat{\mathcal{D}}^c} \mathcal{L}^c(\mathbf{w}^*, \mathcal{D}^c), \quad (6)$$

where $\alpha$ denotes a learning rate for the outer-loop, and the meta knowledge $\hat{\mathcal{D}}^c$ is initialized based on a uniform distribution (*i.e.*, $\hat{\mathcal{D}}^c_{ini} \sim U[-1, +1]$). An updated meta knowledge $\hat{\mathcal{D}}^c$ is obtained in this outer-loop. The inner-loop and outer-loop are conducted in an alternatingly manner.

The presence of data heterogeneity in FL often results in biased meta knowledge extracted on each client. To address this issue, we design two effective mechanisms within our FMKE approach, namely Dynamic Weight Assignment and Meta Knowledge Sharing.

**Dynamic Weight Assignment**: Concretely, we dynamically assign weights for each training sample in $\mathcal{D}$. The prediction confidence of each sample evolves during training, and the dynamic weights is determined based on the varying contributions of each sample to the meta knowledge extraction. To be specific we calculate a dynamic weight value $\phi_i^c$ for each sample $\mathcal{D}_i^c$, *i.e.*, $(x_i^c, y_i^c)$, based on its prediction loss $\ell(\mathbf{w}^c, x_i^c, y_i^c)$. The formulation is defined as follows:

$$\phi_i^c = \frac{1}{1 + \exp\left(-\tau * \ell(\mathbf{w}^c, x_i^c, y_i^c)\right)}, \quad (7)$$

where $\tau$ is a hyper-parameter to smooth the result, $\mathcal{D}^c = \{\mathcal{D}_i^c\}, 1 \le i \le N^c$, and $N^c$ denotes original data number on client $c$. We assign the dynamic weight to each sample to update meta knowledge:

$$\hat{\mathcal{D}}^c \leftarrow \hat{\mathcal{D}}^c - \alpha \nabla_{\hat{\mathcal{D}}^c} \mathcal{L}^c(\mathbf{w}^c, \Phi^c, \mathcal{D}^c), \quad (8)$$

where $\mathcal{L}^c(\mathbf{w}^c, \Phi^c, \mathcal{D}^c) \triangleq \frac{1}{N^c} \sum_{i=1}^{N^c} \phi_i^c \cdot \ell(\mathbf{w}^c, x_i^c, y_i^c)$ [3].

**Meta Knowledge Sharing**: The initialization of $\hat{\mathcal{D}}^c$ is a crucial factor in the extraction of informative meta knowledge. In prior bi-level optimization works (Rajeswaran et al., 2019; Wang et al., 2018), the initialization value was randomly sampled from a constant distribution (*i.e.*, $\hat{\mathcal{D}}^c_{ini} \sim U[-1, +1]$), referred to as unconditional initialization. However, due to the heterogeneity issue in FL, this method may result in the extracted meta knowledge becoming biased towards the corresponding local data. To address this limitation, we propose a *conditional* initialization method incorporating a meta knowledge sharing mechanism.

Conditional initialization (Wang et al., 2020; Denevi et al., 2020) is obtained from data characteristics rather than random generation. To achieve conditional initialization, we design a simple yet effective strategy in extracting meta knowledge $\hat{\mathcal{D}}^c$ for client $c$ at the current round $t$. Specifically, during the initialization process of client $c$, we *randomly* select another client $c'$ and use its meta knowledge $\hat{\mathcal{D}}^{c'}_{t-1}$ extracted in the previous round $t-1$ as the initial value in Eq. 8. This changes the initialization for $\hat{\mathcal{D}}^c$ from an unconditional manner to a conditional manner: $\hat{\mathcal{D}}^c_{ini} \leftarrow \hat{\mathcal{D}}^{c'}_{t-1}, c' \sim randint[1, C]$. In this manner, the meta knowledge condensation for client $c$ is determined by both the local data on client $c$ and the knowledge extracted on another client $c'$, effectively mitigating the heterogeneity issue.

---

[3]For the ease of discussion, $\mathcal{L}^c(.)$ is slightly abused here.

### 2.2.2 SERVER-SIDE CENTRAL MODEL TRAINING

After conducting FMKE, we upload the condensed meta knowledge $\hat{\mathcal{D}}$ from clients to a server. On the server, the uploaded meta knowledge is used as normal training data to train a global model $\mathbf{W}_G$:

$$\mathcal{L}(\mathbf{W}_G, \hat{\mathcal{D}}) = \frac{1}{|\hat{\mathcal{D}}|} \sum_{\hat{x}_i, \hat{y}_i \in \hat{\mathcal{D}}} \ell(\mathbf{W}_G, \hat{x}_i, \hat{y}_i), \tag{9}$$

where $\ell(.)$ is a cross-entropy loss function as in Eq. 2, and $\hat{\mathcal{D}} = \cup \hat{\mathcal{D}}^c, 1 \le c \le C$.

To further ameliorate data biases among diverse clients, we introduce additional synthetic training samples into the central model training. Those introduced training samples are from the same distribution of upload meta knowledge $\hat{\mathcal{D}}$. Specifically, at first, we model the statistical distribution of uploaded meta knowledge $\hat{\mathcal{D}}$ via a conditional generator, and then we sample additional data points based on the learned distribution. Thus, sampled data would share the same distribution as $\hat{\mathcal{D}}$. After the introduction of sampled synthetic data, we not only stabilize our training procedure but also achieve better performance.

To facilitate the discussion, we divide the model $\mathbf{W}_G$ into a feature extractor $\mathcal{F}$ with parameter $\mathbf{W}_G^{\mathcal{F}}$ and a classifier $\mathcal{C}$ with parameter $\mathbf{W}_G^{\mathcal{C}}$, in which $\mathbf{W}_G = (\mathbf{W}_G^{\mathcal{F}}, \mathbf{W}_G^{\mathcal{C}})$. Accordingly, we denote a latent representation as $z = \mathcal{F}(\mathbf{W}_G^{\mathcal{F}}, x)$ and a final prediction as $y' = \mathcal{C}(\mathbf{W}_G^{\mathcal{C}}, z)$. The conditional generator $\mathcal{G}$ maps a label $y$ into a latent representation $z \sim \mathcal{G}(y, \mathbf{w}^{\mathcal{G}})$, and $\mathcal{G}$ is optimized by the objective:

$$\mathcal{G}^* = \arg\max_{\mathcal{G}: y \to z} \mathbb{E}_{y \sim p(y)} \mathbb{E}_{z \sim \mathcal{G}(y, \mathbf{w}^{\mathcal{G}}))} \log p(y|z, \mathbf{W}_G^{\mathcal{C}}), \tag{10}$$

where $\mathbf{w}^{\mathcal{G}}$ denotes the parameter of $\mathcal{G}$.

The trained generator $\mathcal{G}$ models the distribution of uploaded meta knowledge $\hat{\mathcal{D}}$. By sampling data from the distribution, we obtain a set of "pseudo" meta knowledge $\hat{\mathcal{D}}^{pseu}$ with corresponding labels. The generated "pseudo" meta knowledge $\hat{\mathcal{D}}^{pseu}$ as well as uploaded $\hat{\mathcal{D}}$ are utilized to train the global model by minimizing the following function:

$$\mathcal{L}_{overall}(\mathbf{W}_G, \{\hat{\mathcal{D}}, \hat{\mathcal{D}}^{pseu}\}) = \mathcal{L}(\mathbf{W}_G, \hat{\mathcal{D}}) + \beta \mathcal{L}(\mathbf{W}_G, \hat{\mathcal{D}}^{pseu}), \tag{11}$$

where $\beta$ is a parameter and determined by the cardinality fraction $\frac{|\hat{\mathcal{D}}^{pseu}|}{|\hat{\mathcal{D}}|}$.

After central model training, we broadcast the obtained global $\mathbf{W}_G$ and meta knowledge $\hat{\mathcal{D}}$ to clients. On each active client, the broadcasted model $\mathbf{W}_G$ as well as meta knowledge $\hat{\mathcal{D}}$ are used for a new round of FMKE. FMKE and CMT collaborate with each other in an iterative symbiosis paradigm, benefiting each other increasingly as the learning continues. The training process of our FedMK is illustrated in Alg. 1. After the completion of the training process, the trained global model is only used for inference.

**Computational Complexity:** FedMK includes two parts: federated meta knowledge extraction on clients and global model training on the server. On clients, our method adopts a bi-level optimization to extract the meta-knowledge. The bi-level optimization has a running-time complexity of $O(N \times n)$ (Fallah et al., 2020a), in which $n$ denotes the meta knowledge size, $N$ denotes the number of samples on the client. It should be noted that the selected nodes can conduct FMKE in parallel. On the server, the global model training in our method has a running-time complexity of $O(n)$. In total, the overall running-time complexity of our method is $O(N \times n)$.

## 3 EXPERIMENTS

### 3.1 DATASETS

We evaluate our algorithm and compare to the key related works on four benchmarks: MNIST (LeCun et al., 2010), SVHN (Netzer et al., 2011), CIFAR10 (Krizhevsky & Hinton, 2009), and CIFAR100 (Krizhevsky & Hinton, 2009). MNIST is a database of handwritten digits (0-9). In MNIST, there are $50,000$ images in the training set and $10,000$ images in the test set, and their size is $1 \times 28 \times 28$ pixels ($c \times w \times h$). There are 10 classes in MNIST dataset. SVHN is a real-world image dataset, in which there are $600,000$ color images collected from house numbers in Google Street View images. In SVHN, each image is of size $3 \times 32 \times 32$. The class number of SVHN is as the same as MNIST.

---

**Algorithm 1:** FedMK

---

**Input:** Original data $\mathcal{D}$; global parameters $\mathbf{W}_G$; generator parameter $\mathbf{w}^{\mathcal{G}}$; the communication budget.
**Output:** Optimal $\mathbf{W}_G^*$

1 **while** *within the communication budget* **do**
2    the server selects active clients $C$ uniformly at random, and broadcasts $\mathbf{W}_G$ to selected clients $C$.
3    ▷Federated Meta Knowledge Extraction on selected clients $C$:
4    **for** *all user $c \in C$ in parallel* **do**
5       $\mathbf{w}^c \leftarrow \mathbf{W}_G$;
6       **for** *t = 0, ..., #Round* **do**
7          if t == 0: $\hat{\mathcal{D}}_{ini}^c \sim U[-1, +1]$;
8          elif t == 1: conduct the conditional initialization, *i.e.*, $\hat{\mathcal{D}}_{ini}^c \leftarrow \hat{\mathcal{D}}_{t-1}^{c'}, c' \sim randint[1, C]$;
9          calculate dynamic weights by Eq. 7;
10          generate $\hat{\mathcal{D}}_t^c$ by Eq. 4;
11       **end**
12       send the $\hat{\mathcal{D}}_t^c$ to the server.
13    **end**
14    ▷Global Model Training on the server:
15    update generator parameter $\mathbf{w}^{\mathcal{G}}$ by Eq. 10;
16    generate $\hat{\mathcal{D}}^{pseu}$ by the updated generator $\mathcal{G}$;
17    update global parameter $\mathbf{W}_G$ by Eq. 11.
18 **end**
19 return $\mathbf{W}_G$ as $\mathbf{W}_G^*$;

---

**Table 1:** The classification accuracy with limited communication rounds (*i.e.*, 10).

| Setting | FedAvg | FedProx | FedDistill | FedEnsem | FedGen | FedMK |
|---|---|---|---|---|---|---|
| **MNIST** | | | | | | |
| $\alpha$=0.50 | 74.61±2.79% | 73.56±3.84% | 75.04±2.32% | 75.39±1.44% | 74.07±1.20% | **92.95±3.81**% |
| $\alpha$=0.75 | 73.49±3.62% | 73.13±2.68% | 76.21±1.59% | 74.28±1.65% | 74.57±3.98% | **92.86±0.28**% |
| $\alpha$=1.0 | 74.10±1.84% | 73.35±0.10% | 76.19±1.19% | 74.45±2.25% | 73.97±1.69% | **93.63±0.40**% |
| **SVHN** | | | | | | |
| $\alpha$=0.50 | 29.55±2.02% | 28.52±2.44% | 26.92±2.02% | 29.12±2.99% | 28.94±2.62% | **74.11±1.00**% |
| $\alpha$=0.75 | 31.71±0.90% | 25.78±2.76% | 25.77±3.27% | 29.53±1.91% | 30.40±2.86% | **74.90±0.44**% |
| $\alpha$=1.0 | 30.87±1.41% | 29.59±1.26% | 25.84±1.21% | 32.67±2.07% | 33.62±3.28% | **74.84±0.32**% |
| **CIFAR10** | | | | | | |
| $\alpha$=0.50 | 26.63±0.95% | 26.21±0.98% | 24.38±2.12% | 27.56±0.70% | 25.42±2.56% | **47.33±1.08**% |
| $\alpha$=0.75 | 25.42±1.17% | 24.85±1.41% | 24.18±1.25% | 26.42±0.59% | 26.25±0.33% | **49.04±1.15**% |
| $\alpha$=1.0 | 26.80±0.89% | 26.66±0.37% | 25.83±0.79% | 26.74±0.87% | 25.36±1.32% | **50.32±0.69**% |
| **CIFAR100** | | | | | | |
| $\alpha$=0.50 | 11.66±1.22% | 12.09±0.72% | 10.76±0.77% | 13.20±0.44% | 10.34±0.65% | **26.74±0.60**% |
| $\alpha$=0.75 | 12.11±0.45% | 11.65±0.36% | 11.55±0.79% | 13.15±0.20% | 10.21±1.02% | **27.43±0.54**% |
| $\alpha$=1.0 | 12.31±0.60% | 11.34±0.48% | 11.50±0.66% | 13.31±0.54% | 11.19±0.75% | **28.20±0.10**% |

CIFAR10 dataset consists of $60,000$ color images, each of which has a size of $3 \times 32 \times 32$. There are $50,000$ images in the training set and $10,000$ images in the testing set. There are 10 classes in CIFAR10. CIFAR100 dataset consists of $60,000$ color images, each of which has a size of $3 \times 32 \times 32$. There are $50,000$ images in the training set and $10,000$ images in the testing set. For each image in CIFAR100, there are two kinds of labels, *i.e.*, fine label and coarse label. We choose coarse labels and therefore we have 20 classes in the experiment on CIFAR100.

### 3.2 IMPLEMENTATION DETAILS

We set the user number to 20, and the active-user number to 10. We use $50\%$ of the training set and distribute it on all clients. All testing data is utilized for evaluation. We use LeNet (LeCun et al., 1989) as the backbone for all methods: FedAvg (McMahan et al., 2017), FedProx (Li et al., 2020), FedDistill (Seo et al., 2020), FedEnsemble (Zhu et al., 2021), FedGen (Zhu et al., 2021), and FedMK. Dirichlet distribution $Dir(\alpha)$ is used to model data distributions. Specifically, we test three different $\alpha$ values: $0.5$, $0.75$, and $1.0$, respectively. We set the communication round number to 10. Learning with a small communication round number (*e.g.*, 10) denotes learning under a limited communication budget. For the methods conducting local model training on clients, *i.e.*, FedAvg, FedProx, FedDistill, FedEnsemble, and FedGen, we set the local updating number to 20, and the batch size number to 32. In our method, we set meta knowledge size for each datasets based on their different characteristics (*e.g.*, the class number, sample size, etc): 20 per class for MNIST; 100 per class for SVHN; 100 per

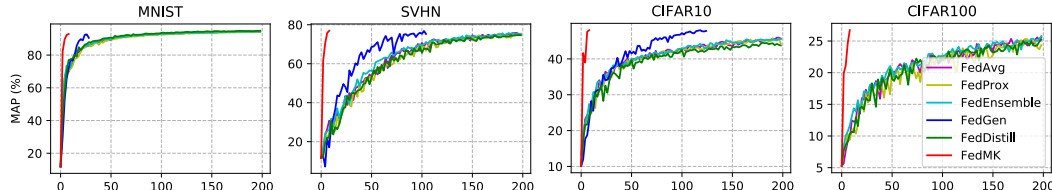

**Figure 2:** Convergence rate comparisons on MNIST, SVHN, CIFAR10, and CIFAR100. $\alpha = 1.0$. The x-axis represents communication round numbers.

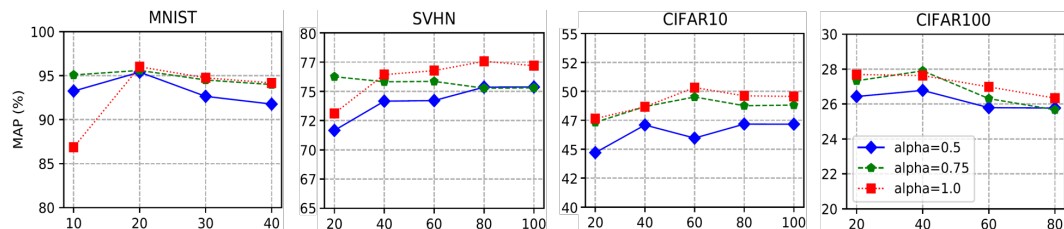

**Figure 3:** Impact of meta knowledge size on final performance. #round=10. The x-axis represents meta knowledge sizes.

**Table 2:** Impact of each designed mechanism.

|          | w/o Iter | w/o Sharing | w/o pseudo knowledge | w/o dynamic weights | Ours    |
|----------|----------|-------------|----------------------|---------------------|---------|
| CIFAR10  | 28.15%   | 45.79%      | 46.71%               | 47.16%              | **47.33%** |
| CIFAR100 | 18.82%   | 25.56%      | 26.10%               | 26.43%              | **26.74%** |

class for CIFAR10; 40 per class for CIFAR100. We run three trials and report the mean accuracy performance (MAP).

### 3.3 COMPARATIVE STUDIES

**Compared with Prior Works:** We evaluate FedMK and compare it with relevant prior works under limited communication budgets (10 rounds) on four datasets, including MNIST (LeCun et al., 2010), SVHN (Netzer et al., 2011), CIFAR10 (Krizhevsky & Hinton, 2009), and CIFAR100 (Krizhevsky & Hinton, 2009). The results are reported in Table 1. As shown in the results, under limited communication budgets, all prior works fail to achieve good performance, while FedMK outperforms them by a significant margin. For instance, on MNIST, FedMK achieves a MAP of 93.63% ($\alpha = 1$), outperforming the competing methods. On SVHN, FedMK reaches a MAP of 75.85% ($\alpha = 1$) in 10 rounds, while the competing methods have not converged yet. The superior performance of FedMK can be attributed to two aspects. Firstly, unlike competing methods that conduct local model training with local private data, FedMK trains a global model based on meta-knowledge from all active clients, which reduces the bias in the model. Secondly, the dynamic weight assignment and meta knowledge sharing mechanisms designed in FedMK make the meta knowledge extraction more stable, leading to better performance than competing methods [4].

**Convergence rate comparisons:** We compare the convergence rates of all methods and show the performance curves with respect to communication rounds in Figure 2. As expected, our method achieves a high convergence speed compared to all competing methods. On all datasets, our method achieves satisfactory performance in much fewer communication rounds. The results shown in both Table 1 and Figure 2 demonstrate the effectiveness and efficiency of our proposed method.

**Impact of the meta knowledge size:** We study the effect of varying meta knowledge sizes on the performance of our method on four benchmark datasets. The results are presented in Figure 3. The results show that the final performance is influenced by the meta knowledge size. For instance, the MAP score on CIFAR10 improves as the meta knowledge size increases. Furthermore, we find that the optimal meta knowledge sizes can differ between datasets. For example, the highest MAP score on MNIST is achieved when the meta knowledge size is set to 20, while the optimal meta knowledge sizes on the other three datasets are different. Further discussions on the relationship between meta knowledge size, communication rounds, and performance can be found in the Appendix.

---

[4]A more detailed comparison with FedGen can be found in the Appendix.

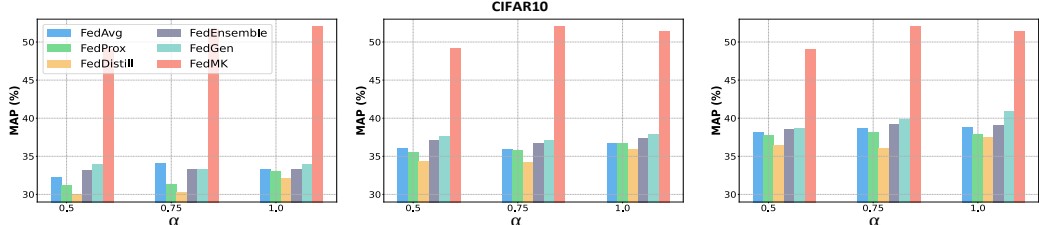

**Figure 4:** The performance curve w.r.t. the active client number. $\alpha = 0.5$, #node = 20, #round = 10. The x-axis represents the active client numbers.

**Figure 5:** Performance comparisons w.r.t. different round numbers. Left column: 20 rounds; middle column: 30 rounds; right column: 40 rounds. The x-axis represents different $\alpha$ values.

**Impact of active client numbers**: We examine the effect of varying the number of active clients on our proposed method and competing works. We conduct experiments with active client numbers set to 5, 7, and 10, respectively. The performance results are displayed in Figure 4. The results illustrate that our proposed method consistently outperforms all comparison methods regardless of the number of active clients.

**Impact of communication rounds:** We conduct an experiment on the CIFAR10 dataset to analyze the impact of communication rounds. We compare the performance of different methods when the number of communication rounds is set to 20, 30, and 40. The results of the comparison are presented in Figure 5. The results demonstrate that our proposed method consistently outperforms the competing methods by a remarkable margin.

**Impact of designed mechanisms**: We conduct a study on CIFAR10 and CIFAR100 to explore the impact of each designed mechanism. We report the performance in Table 2, where "w/o Iter" means that we conduct local meta knowledge condensation and global model update only once [5], "w/o Sharing" means that we do not adopt meta knowledge sharing between clients, "w/o pseudo meta knowledge" means that there is no learnable constraint on the server, "w/o dynamic weights" means that we update $\hat{\mathcal{D}}$ by treating all samples in $\mathcal{D}$ equally.

As shown in Table 2, there is a drastic performance degradation when there is no iterative collaboration between meta knowledge extraction and central model training ("w/o Iter"). Making the meta knowledge condensation and central model training running in an iterative manner improves the performance significantly, *e.g.*, on CIFAR10, MAP score increases from 28.15% to 45.79%. Moreover, the two designed mechanisms, *i.e.*, the meta knowledge sharing between clients, dynamic weights assignment, significantly boost the MAP score.

**Evaluation on the pathological non-iid setting:** We conduct an experiment on MNIST under the pathological non-iid setting (Huang et al., 2021). In the experiment, we set the node number to 20, the active node number to 10, the number of classes on each client to 5, $\alpha$ to 1.0. We compare all methods under a limited budget communication (*i.e.*, 10 rounds). As seen in Table 3, our method achieves better performance compared to the competing methods.

**The privacy concerns of meta knowledge:** Dong et al. (2022) conducted a theoretical analysis of the relationship between synthetic data and data privacy. Based on Proposition 4.10 in (Dong et al., 2022), when using distilled data (meta knowledge), the leakage of membership privacy that directly relates to personal privacy is $O(\frac{card(\hat{\mathcal{D}})}{card(\mathcal{D})})$. As indicated by Proposition 4.10, when the cardinality

---

[5]"w/o Iter" can be regarded as one-shot FL like Zhou et al. (2020a)

**Table 3:** Result comparisons on MNIST under the pathological non-iid setting.

|  | FedAvg | FedProx | FedEnsem | FedDistill | FedGen | Ours |
|---|---|---|---|---|---|---|
| **MAP (%)** | 75.11% | 72.99% | 76.43% | 71.19% | 86.44% | **88.85**% |

of the synthetic data is much fewer than that of the original data, only limited (i.e., $O(\frac{card(\hat{\mathcal{D}})}{card(\mathcal{D})})$) information can be obtained by the adversary with membership inference attack.

In the Appendix, we provide more experimental analysis and comparisons with prior works.

## 4 RELATED WORK

**Federated Learning.** As a privacy-preserving solution, federated learning (Li et al., 2019) provides a new training manner to learn models over a collection of distributed devices. In federated learning, the data is located on distributed nodes and not shared. Learning a model without exchanging local data between nodes minimizes the risk of data leakage but increases the difficulty of training. To address this issue, FedAvg (McMahan et al., 2017) was proposed to obtain a global model by aggregating local models trained on active clients. To further enhance local training, personalized FL methods (Fallah et al., 2020b; Dinh et al., 2020; Dai et al., 2022) and knowledge distillation-based FL methods (Seo et al., 2020; Zhu et al., 2021; Zhang et al., 2022) have been proposed. For a comprehensive understanding of FL, readers are referred to (Kairouz et al., 2021; Tan et al., 2022; Gao et al., 2022).

**Compact Data Representation.** Generally, prior works compressing a large scale data into a small set can be categorized into two main branches: data selection and data compression. Data selection methods (Rebuffi et al., 2017; Castro et al., 2018; Aljundi et al., 2019; Sener & Savarese, 2018) select the most representative samples from the original data based on predefined criteria. How to select an appropriate criterion based on the given data and task is not a trivial issue. To overcome the aforementioned limitations, synthesizing new samples rather than selecting existing samples becomes a more preferable solution. The methods (Wang et al., 2018; Zhao et al., 2021; Nguyen et al., 2021a; Zhao & Bilen, 2021; Nguyen et al., 2021b; Vicol et al., 2022; Cazenavette et al., 2022; Zhou et al., 2022; Loo et al., 2022; Wang et al., 2022; Lee et al., 2022; Jiang et al., 2022; Du et al., 2022; Zhao & Bilen, 2023; Loo et al., 2023) design different solutions for generating synthetic data based on given datasets. In those methods, the generated data can replace the original data in the model construction process. However, those prior synthetic data generation works require data to be localized in a centralized manner.

**Federated Learning via Synthetic Data.** A few attempts (Goetz & Tewari, 2020; Zhou et al., 2020b; Yoon et al., 2021; Xiong et al., 2022; Hu et al., 2022; Kim & Choi, 2022; Behera et al., 2022; Song et al., 2022) have employed synthetic data generated from local data for FL. Zhou et al. (2020b) and Song et al. (2022) introduced one-shot FL methods by utilizing distilled data. Goetz & Tewari (2020) proposed a method for FL that utilizes data-poisoning to generate synthetic training data. Behera et al. (2022) utilized synthetic data generated by a Generative Adversarial Network for FL training. Concurrently, Xiong et al. (2022) used gradient matching to synthesize data from the original data and used the synthetic data to train a global model. Hu et al. (2022) and Kim & Choi (2022) also adopted the distilled data for training a global model in FL. Compared to these concurrent works (Xiong et al., 2022; Hu et al., 2022; Kim & Choi, 2022), our method introduces a meta-knowledge sharing mechanism and a dynamic weight assignment strategy, significantly increasing the informative content of the meta knowledge. In this way, we can further speed up the convergence of the global model. In FedMix (Yoon et al., 2021), Yoon *et al.* proposed to share the averaged local data among clients for knowledge sharing. However, FedMix (Yoon et al., 2021) still follows the training paradigm of FedAvg (*i.e.*, training local models on clients and aggregating a global model on the server), incurring a higher communication cost.

## 5 CONCLUSION

In this paper, we present a new federated learning paradigm driven by meta knowledge, dubbed FedMK, to obtain an effective and fast-converging model. With the help of the proposed paradigm, FedMK can train a powerful model even under a limited communication budget (*e.g.*, 10 communication rounds), decreasing the communication cost significantly. Moreover, our designed mechanisms, *i.e.*, meta knowledge sharing, dynamic weight assignment, and a learned constraint, collectively facilitate the central model training, benefiting FedMK outperforming all competing methods.

## ACKNOWLEDGEMENT

This work was supported by A*STAR Career Development Funding (CDF) Award (Grant No: 222D800031), SERC (Science and Engineering Research Council) Central Research Fund (Use-Inspired Basic Research), and in part by the Singapore Government's Research, and Innovation and Enterprise 2020 Plan (Advanced Manufacturing and Engineering Domain) under programmatic Grant A18A1b0045. Xin YU was supported by the Australian Research Council Discovery Project (DP220100800) and DECRA (DE230100477).

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

**Table 4:** Results with 10 rounds.

| Setting | FedAvg | FedProx | FedDistill | FedEnsem | FedGen | FedMK |
|---------|--------|---------|------------|----------|--------|-------|
| **MNIST** | | | | | | |
| $\alpha$=0.10 | 61.95% | 61.41% | 58.46% | 67.89% | 64.83% | **77.37%** |
| $\alpha$=0.25 | 69.52% | 68.43% | 71.78% | 72.23% | 73.41% | **90.07%** |
| **SVHN** | | | | | | |
| $\alpha$=0.10 | 20.10% | 18.39% | 25.44% | 24.60% | 24.38% | **57.24%** |
| $\alpha$=0.25 | 23.56% | 25.01% | 22.70% | 23.21% | 28.79% | **65.66%** |
| **CIFAR10** | | | | | | |
| $\alpha$=0.10 | 23.71% | 21.88% | 24.93% | 24.80% | 20.16% | **38.45%** |
| $\alpha$=0.25 | 21.85% | 22.17% | 20.84% | 23.98% | 22.94% | **40.75%** |
| **CIFAR100** | | | | | | |
| $\alpha$=0.10 | 10.19% | 9.41% | 12.41% | 10.64% | 10.79% | **18.62%** |
| $\alpha$=0.25 | 11.73% | 10.43% | 8.73% | 12.42% | 8.22% | **22.14%** |

**Appendix Meta Knowledge Condensation for Federated Learning**

## A    ADDITIONAL EXPERIMENTAL RESULTS

**Compared with Prior Works when setting $\alpha$ to 0.1 and 0.25:** We set the $\alpha$ value in Dirichlet distribution $D(\alpha)$ (Zhu et al., 2021) to 0.1 and 0.25, and run all methods under limited communication budgets (10 rounds) on four datasets. We report the results in Table 4. As shown in Table 4, when communication budgets are limited (10 rounds) and the $\alpha$ value is set to 0.25 and 0.10, our method can still learn a model outperforming competing works by a remarkable margin.

**Impact of the weight assignment mechanism:**  We investigate the impact of weight assignment mechanisms in our proposed method. Here, we compare our Dynamic Weight Assignment mechanism with a Class Balance Weight strategy that has been used in prior works (Tian et al., 2022; Shao et al., 2021). This strategy assigns weights to each sample during training based on the occurrence rate of the corresponding class. We perform an experiment on the MNIST dataset and present the results in Table 6. The results demonstrate that our proposed Dynamic Weight Assignment mechanism outperforms the Class Balance Weight strategy.

**Impact of node numbers:** We conduct experiments on the SVHN dataset to analyze the effect of the number of nodes. Specifically, we set the node number to 200 for comparisons. Typically, as the number of nodes increases, the amount of data on each client becomes sparse, making learning more challenging and leading to a decrease in performance. However, our experiment shows that even when the node number is increased to 200, our method (70.51%) still outperforms FedGen (66.31%) by a substantial margin.

**Impact of the smooth parameter $\tau$ in Eq. 7:** We conduct a study on four datasets to explore the impact of smooth parameter $\tau$ in Eq. 7. We set the $\tau$ value to 1.0, 5.0, and 10.0, respectively. As shown in Table 5, we achieve the highest performance in most cases when we set $\tau$ to 5.0. Therefore, we set $\tau$ in Eq. 7 to 5.0 for all our experiments.

**Communication cost comparisons:** We analyze the communication cost of our method on MNIST. In each round of FedMK, we need to upload and download the meta knowledge, and download a trained model. In the experiment, the meta image size is 20 per class, each of which is 4 Bytes $\times$ $28 \times 28$ (sizeof(float32)$\times w \times h$). Thus, in each round, the uploading cost of FedMK is 627.2K (meta knowledge), *i.e.*, 4 Bytes $\times$ $28 \times 28 \times 20 \times 10$, and the downloading cost is 1677.2K (meta knowledge and a model), *i.e.*, 627.2K + 105K $\times 10$. In total, the communication cost for FedMK is 16M, *i.e.*, $(627.2K+1,677.2K) \times 10$ (# rounds). As shown in Figure 2, to achieve comparable performance on MNIST, FedAvg has to run around 200 rounds. In this case, the communication cost for FedAvg is 420M, *i.e.*, $105K \times 10 \times 2 \times 200$ (model size $\times$ #active node $\times$ 2 $\times$ #communication). This indicates that to obtain models with good performance, FedAvg and its variants require much higher communication costs.

**Table 5:** Impact of the smooth parameter $\tau$.

|  | $\tau$=1.0 | $\tau$=5.0 | $\tau$=10.0 |
|---|---|---|---|
| **MNIST** | | | |
| $\alpha$=0.50 | 91.70% | **92.95%** | 91.79% |
| $\alpha$=0.75 | 91.90% | **92.86%** | 92.23% |
| $\alpha$=1.0 | 91.53% | **93.63%** | 91.91% |
| **SVHN** | | | |
| $\alpha$=0.50 | 72.24% | **74.11%** | 71.60% |
| $\alpha$=0.75 | 71.47% | **74.90%** | 74.03% |
| $\alpha$=1.0 | 71.74% | **74.84%** | 72.19% |
| **CIFAR10** | | | |
| $\alpha$=0.50 | **47.72%** | 47.33% | 46.62% |
| $\alpha$=0.75 | 48.17% | 49.04% | **49.27%** |
| $\alpha$=1.0 | 47.82% | **50.32%** | 48.54% |
| **CIFAR100** | | | |
| $\alpha$=0.50 | 26.06% | **26.74%** | 26.45% |
| $\alpha$=0.75 | 26.93% | **27.43%** | 26.98% |
| $\alpha$=1.0 | 25.43% | **28.20%** | 26.15% |

**Table 6:** Comparisons on MNIST under different weight assignment mechanisms.

| Method | $\alpha = 0.5$ | $\alpha = 0.75$ | $\alpha = 1.0$ |
|---|---|---|---|
| Balance Class Weight | 92.17% | 91.84% | 91.69% |
| **Our method** | **92.95%** | **92.86%** | **93.63%** |

**Table 7:** EMDs with respect to meta-knowledge sizes.

| Meta-Knowledge size(S) | 10 | 20 | 30 | 40 | 50 | 60 | 70 | 80 |
|---|---|---|---|---|---|---|---|---|
| **Difference w.r.t. S=10** | 0 | 10 | 20 | 30 | 40 | 50 | 60 | 70 |
| **EMD**($meta_{\mathbf{S}}, meta_{10}$) | 0.000 | 0.023 | 0.054 | 0.046 | 0.053 | 0.045 | 0.045 | 0.043 |

# B    DISCUSSIONS

**Why does utilizing meta-knowledge decrease the required communication rounds?** Our method utilizes extracted meta knowledge as normal training data to train a global model on the server. The meta knowledge is extracted from original data via a bi-level optimization, which encodes the "gradient of gradient" with respect to the model. The optimization methods based on the second order gradient generally have a higher convergence speed than the methods using the first order gradient (Battiti, 1992; Xie et al., 2022). Therefore, utilizing meta-knowledge endows our algorithm with a fast convergence speed and decreases the communication round number.

**Why increasing the meta knowledge size can not necessarily improve final performance?** In the meta knowledge extraction process, the calculated meta-knowledge in each batch represents the average of the model update direction. The average gradient is stable when the batch number increases in a certain range (from ×1 to ×10). As a result, increasing the meta-knowledge sizes does not necessarily increase the performance. Intuitively, the meta knowledge is highly dense and compressed, encoding the knowledge from original data (Zhou et al., 2022). In principle, using the meta knowledge approximates employing the original data. As the amount of information in the original data is constant, the training performance will not necessarily increase as the meta knowledge size increases.

We conduct an experiment on MNIST to show the information change between meta knowledge with respect to different sizes. Concretely, we set the meta-knowledge size (S) as 10, 20, 30, 40, 50, 60, 70, and 80, respectively. As earth mover's distance (EMD) has been utilized to compute a structural distance between two data sets to determine their similarity (Zhang et al., 2020), we use it to evaluate differences according to meta-knowledge with different sizes. The results are listed in Table 7. It can be seen that the EMDs with respect to meta-knowledge sizes are stable, indicating the amount of information in the meta-knowledge does not change significantly with respect to the meta knowledge size.

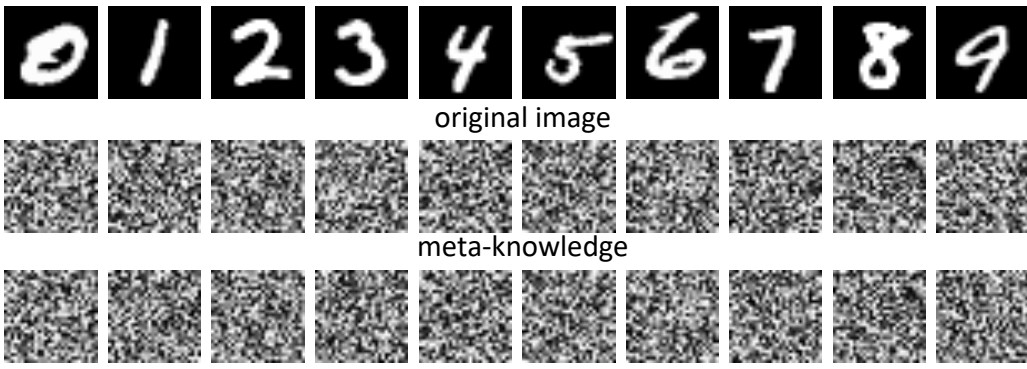

**Figure 6:** The visualization of original images (the top row), extracted meta-knowledge (the middle row), and restored data by deep leakage based on extracted meta-knowledge (the bottom row).

**The privacy concerns of meta knowledge.** We think that the use of generated meta knowledge can protect original data privacy. This can be argued from three perspectives. Firstly, as shown in the function for the upper-level problem, *i.e.*, $\hat{\mathcal{D}}^c \leftarrow \hat{\mathcal{D}}^c - \alpha \nabla_{\hat{\mathcal{D}}^c} \mathcal{L}^c(\mathbf{w}^*, \mathcal{D}^c)$, it never returns the original local data $\mathcal{D}^c$. What we receive in the outer loop (the upper-level problem) is the updated (or initialized when there is no update) meta knowledge $\hat{\mathcal{D}}^c$.

Secondly, based on Deep Leakage (Zhu & Han, 2020), we conduct an experiment on MNIST to explore the possibility of restoring data from extracted meta knowledge. The results are shown in Figure 6. The original images are shown in the top row, and the extracted meta knowledge is shown in the middle row. We feed the extracted meta knowledge and a trained model to Deep Leakage (Zhu & Han, 2020), which is one of the state-of-the-art methods for restoring data from leaked knowledge. The data restored by Deep Leakage is shown in the bottom row. It can be seen that it is hard to construct correspondence between entries in restored data and original data.

Thirdly, Dong *et al.* (Dong et al., 2022) conducted a theoretical analysis of the relationship between synthetic data and data privacy. Based on Proposition 4.10 in (Dong et al., 2022), when using distilled data (meta knowledge), the leakage of membership privacy that directly relates to personal privacy is $O(\frac{card(\hat{\mathcal{D}})}{card(\mathcal{D})})$. As indicated by Proposition 4.10, when the cardinality of the synthetic data is much fewer than that of the original data, only limited (i.e., $O(\frac{card(\hat{\mathcal{D}})}{card(\mathcal{D})})$) information can be obtained by the adversary with membership inference attack.

## C  THE COMPARISON WITH PRIOR WORKS

**The difference between FedMix (Yoon et al., 2021) and FedMK:** Our method differs from FedMix (Yoon et al., 2021) significantly in three aspects.

Firstly, our method adopts a different training paradigm. FedMix (Yoon et al., 2021) follows the training paradigm of FedAvg, i.e., training local models on clients and aggregating a global model on the server. Our method conducts the meta knowledge extraction on clients, and trains a global model on the server based on the uploaded meta knowledge. Our method significantly decreases the communication cost as it only needs to upload the meta knowledge.

Secondly, our method has a faster convergence speed. This is because our method utilizes the second-order gradient information. In FedMix (Yoon et al., 2021), only the first-order gradient information is utilized. In our method, the meta knowledge encodes the "gradient of gradient" with respect to the model. The optimization methods based on the second-order gradient generally have a higher convergence speed than the methods using the first-order gradient (Xie et al., 2022). We conduct an experiment on MNIST ($\alpha = 1.0$) to compare with FedMix (Yoon et al., 2021). When communication

budgets are limited (i.e., 10 rounds), FedMix only achieves $74.68\%$ MAP, whereas our method can achieve $93.63\%$ MAP.

Thirdly, to ameliorate the heterogeneity issue in FL, we specifically design two mechanisms in federated meta knowledge extraction on local clients, i.e., dynamic weight assignment and meta knowledge sharing. There are no such mechanisms in FedMix (Yoon et al., 2021). We demonstrate that the two mechanisms can improve the final accuracy significantly (*e.g.*, 1.5% improvement on CIFAR10).

In summary, compared to FedMix (Yoon et al., 2021), our method adopts a different FL training paradigm, utilizes the second-order gradient information in model learning, and designs two specific mechanisms for ameliorating the heterogeneity issue.

**The difference between FedSynth (Hu et al., 2022), FedDM (Xiong et al., 2022), FedDDC (Kim & Choi, 2022) and FedMK:** Although our work and FedSynth (Hu et al., 2022), FedDM (Xiong et al., 2022), FedDDC (Kim & Choi, 2022) all use condensed data in FL, there are significant differences between them. Please allow us to illustrate the differences from three aspects.

Firstly, on local clients, our method designs a different federated meta knowledge extraction solution. Prior works such as (Xiong et al., 2022; Kim & Choi, 2022) directly utilize a gradient matching strategy for data condensation. We adopt a bi-level optimization solution in our method. To further ameliorate the heterogeneity issue in FL, we specifically design two mechanisms in our meta knowledge extraction process, *i.e.*, dynamic weight assignment and meta knowledge sharing. There are no such mechanisms in prior works. In our experiment, we demonstrate the effectiveness of the two mechanisms (*e.g.*, 1.5% improvement on CIFAR10).

Secondly, on the server, prior works only use the uploaded condensed data to finetune the aggregated global model. In our method, to further ameliorate data biases among diverse clients, we introduce additional synthetic training samples generated by a conditional generator in the global model training.

Last but not the least, our method adopts a different FL learning paradigm compared to FedDC (Kim & Choi, 2022). Kim et al. (Kim & Choi, 2022) conduct local model training on clients, global model aggregation and fine-tuning on the server. Our method conducts meta knowledge extraction on clients, global model training on the server. As our method does not need to upload the local trained model in each communication round, our method significantly decreases the communication cost.

In summary, there are significant differences between prior works and our proposed method in various aspects.

**The difference between FedGen (Zhu et al., 2021) and FedMK:** There are significant differences between FedGen (Zhu et al., 2021) and our FedMK, which are listed as follows:

Firstly, FedGen utilizes original data on local clients to train local models, while our method conducts meta knowledge extraction to synthesize meta knowledge, which is used for global model training on a server. In FedGen, the trained local models might diverge due to the data distribution variations among clients.

Secondly, FedGen constructs a global model by aggregating uploaded local models; while our method learns a global model based on meta knowledge uploaded from clients. In our method, the global model learning utilizes knowledge from all active clients, therefore mitigating the bias issue compared to FedGen.

Thirdly, the conditional generator in FedGen is trained on the server and transmitted to clients. On clients, it is used as a constraint in the local model training. On the contrary, the conditional generator in our method is trained and utilized on the server, participating in the global model training. Compared to FedGen, our method has a less communication cost without performance deterioration.

In conclusion, compared to FedGen, our method performs more effectively and efficiently.

