# OpenReview forum: "Meta Knowledge Condensation for Federated Learning"
_ICLR.cc/2023/Conference — ICLR 2023 poster_

### Official Review · Reviewer_FoXU · 2022-10-24

**Confidence:** 3
**Correctness:** 2
**Technical Novelty And Significance:** 3
**Empirical Novelty And Significance:** 3
**Recommendation:** 6

**Clarity, Quality, Novelty And Reproducibility:**

Clarity: Idea is generally clear. As mentioned, eq. (3) is confusing and I will need clarification from the authors.
Quality: I currently think the approach is not sound. I hope the author can clarify if I have misunderstood anything. Despite my doubt, the empirical result seems great.
Novelty: Most components are applications of previous work (as acknowledged by the authors), but the overall idea is quite novel.
Reproducibility: No code was provided in the supplementary material. No anonymous github page was provided.

**Strength And Weaknesses:**

Strength: Idea is clear and straight forward. The empirical result seems great.
Weakness: The objective formulation seems problematic and I cannot make sense why it works in practice.

-- The notation in Eq. (3) does not make sense to me. $D_c$ is a constant, not a variable. Why is it being optimized? I take it as this is some random variable $D'$ that has the same desired dimension as the condensed dataset. If so, how is the local dataset used?

-- Another major concern is that Eq. (3) seems to have a trivial solution? First off, i think it is improper to have loopy dependency in an optimization objective ($\widehat{D}_c$ depends on $w_\ast$ and vice versa). Maybe it's better to write it as $\widehat{D}_c, w_\ast = \underset{w_c, D_c}{\mathrm{argmin}} \dots $, i.e., the bilevel optimization is an approximate procedure to solve for this objective and should not be the objective itself. Regardless of the notation, I think the optimization will converge when $\widehat{D}_c$  and $w_\ast$ are obtained such that $\mathcal{L}(\widehat{D}_c, w_\ast) = 0$. This is probably simple to achieve when there is no constrained on the meta dataset. For example, we can construct a dataset where all datapoints are labelled as class 1, and a model that always predicts class 1 with zero uncertainty (setting the bias of the prediction layer to $\mathbf{e}_1$ and zeroing out all other weights would probably suffice). Thus, a trivial solution exists and theoretically this could result in very bad performance. This seems intriguing given the very good result observed in the empirical studies. My best explanation for this good performance is that the local models only perform their bilevel optimization for a few steps, which is likely not enough to find such an exploit.

-- This objective does not seem to explicitly prevent the condensed dataset from being exactly the same as the local data. After all, they live in the same input space, so it could happen. I understand that the paper conducts an empirical study to verify this is not the case, but when you deal with privacy, that is simply not good enough. What will happen when the global model somehow has zero loss on the client data? Then, the outer loop doesn't need to do anything but returning a subset of its local data (since loss can't be improved further). This seems like a big privacy vulnerability to me --- Imagine a client with only 10 data points on a binary classification task, then a malicious server would only need to send $2^10$ specifically constructed weights to eventually hit the perfect loss and hence recover exactly the client data? I think the meta dataset somehow has to have a different dimension than the original dataset for this to work.

Some other (minor) concerns:

--  We run three trials and report the mean accuracy performance (MAP) -- is the MAP averaged over both trials and clients? I would recommend showing some standard deviations.

-- How many update steps per round of local data distillation?

-- How is evaluation done? Does each local model of FMKE fine-tune the global weights before testing on their own data, or do they directly use the global weights trained on synthetic data? Is it the same for other benchmarks?

-- Can the authors show the MAP curve of FMKE beyond iteration 10 in Fig. 2? It's not just about achieving high performance, but also about demonstrating the stability of the method.

**Summary Of The Paper:**

This paper proposes a new approach for Federated learning. The idea is to first distill the private data of each client (with respect to the current initialization of their model weights) and send them to the server. On the server side, a global model will learn from these synthetic data and broadcast the learned weight back to the clients. The proposed method is demonstrated to outperform other FL benchmarks on several datasets.

**Summary Of The Review:**

I recommend a rejection since I am not fully convinced about the distillation objective. I'm willing to change my score if the authors can clarify my concerns. I'm personally very curious about the exceptionally good results. I do hope that my assessment is wrong, because the method otherwise seems like a good contribution. I would suggest the authors to release the implementation on an anonymized github repo, so reviewers can properly investigate.

-------

Upon reading the authors' responses to my questions and revisions of the manuscript, I am convinced that the method is sound (Another minor comment: The revision is technically not a bilevel optimization anymore since both parameter updates happen simultaneously). As promised, I'm happy to change my scores accordingly.

---

### Official Review · Reviewer_StBv · 2022-10-24

**Confidence:** 4
**Correctness:** 3
**Technical Novelty And Significance:** 3
**Empirical Novelty And Significance:** 3
**Recommendation:** 6

**Clarity, Quality, Novelty And Reproducibility:**

Clarity: Overall, the main idea, methodology, and the key experimental results are present clearly, except for some issues that could be further improved as mentioned above.

Quality: Goodness. The paper presents an effective solution to deal with an important problem.

Novelty: The core idea of the paper is novel, while some of the techniques are based on or similar to existing solutions.

Reproducibility: The experimental settings are clarified in this paper, but it would be better if the source code is made publicly available.


**Strength And Weaknesses:**

Strength:
S1: A new FL paradigm is proposed to deal with the limited communication budgets problem by leveraging condensate meta knowledge instead of the raw training data.
S2: The proposed paradigm is significantly better than existing FL solutions when facing limited communication rounds.
S3: Extensive experiments are conducted to show the effectiveness of the proposed method in terms of accuracy, communication efficiency (rounds) and also data leakage.

Weaknesses:
W1: I wonder if it is fair to evaluate the communication budgets using the metric of communication rounds, because unlike the competitors that the clients and the servers transmit the gradients or model weights, the FMKE framework proposed in this paper needs the clients to send the meta data, which have different amount of size.
W2: It is unknown if the bi-level optimization is optimal for FMKE framework. It is expected to compare the bi-level optimization with other baselines, e.g., simultaneously optimizing the two objectives.
W3: While dynamically adjusting the weights of the training samples has been widely studied, it lacks of comparisons between the proposed Dynamic Weight Assignment with the existing methods, such as AdaBoost.


**Summary Of The Paper:**

This paper proposes a new meta-knowledge driven FL paradigm to optimize the FL training with limited communication rounds. Besides, two strategies namely Dynamic Weight Assignment and Meta Knowledge Sharing are proposed to improve the performance.

**Summary Of The Review:**

Overall, the paper presents a new solution to tackle the FL training problem in case of limited communication rounds. The core performance is well evaluated. However, there still remains some issues that could be further improved. Thus, I recommend this paper a borderline accept score.

---

### Official Review · Reviewer_QKpf · 2022-10-25

**Confidence:** 4
**Correctness:** 4
**Technical Novelty And Significance:** 4
**Empirical Novelty And Significance:** 4
**Recommendation:** 8

**Clarity, Quality, Novelty And Reproducibility:**

Clarity: The paper is well-written and easy-to-follow.
Quality: The technical quality is good. The method proposed in this paper is extensively evaluated.
Novelty: The idea of this paper is novel. I haven't seen the same idea in prior works. Reproducibility: This work is with good reproducibility.

**Strength And Weaknesses:**

Strength:

-- The authors designed a new learning solution. Unlike prior FedAvg-based methods, the authors conduct meta knowledge extraction and upload the meta knowledge to a server. They use the meta knowledge from all active clients as normal training data for training a global model, mitigating the bias issue. The meta knowledge extraction and global model training are conducted alternatively.

-- The designed mechanisms, i.e., meta knowledge sharing and dynamic weight assignment, are novel and technically correct.

-- The authors conduct extensive experiments and ablation studies, and provide convincing experimental results to demonstrate the efficacy and efficiency of their method.

-- The writing is clear and easy to understand. The whole work is organized well.

Weakness:
Two minor issues:
-- It would be better to place the Algorithm pseudo code in the draft rather than supplementary material.
-- Why name the extracted knowledge as meta knowledge? Is there any specific reason? If yes, it would be better to provide the reason in the paper.

Clarity, quality, novelty and reproducibility
Clarity: The paper is well-written and easy-to-follow.
Quality: The technical quality is good. The method proposed in this paper is extensively evaluated. Novelty: The idea of this paper is novel. I haven't seen the same idea in prior works. Reproducibility: This work is with good reproducibility.

**Summary Of The Paper:**

This paper develops a new FL solution. In the designed solution, the local clients distill meta knowledge based on local private data and shared knowledge from other clients, the distilled meta knowledge is uploaded to the server for global model training. As the global model training is based on meta knowledge from all active clients, the bias issue can be mitigated and the convergence can be accelerated experimentally. Additionally, the authors designed two novel mechanisms to improve the meta knowledge extraction on local clients, i.e., meta knowledge sharing and dynamic weight assignment. Both of the designed mechanisms are novel and technically correct. The authors conduct extensive experiments and ablation studies to demonstrate the efficacy and efficiency of their method.

**Summary Of The Review:**

This work is interesting and novel. The authors designed a new learning paradigm and designed two novel mechanisms in the learning. The motivation is explained well and the technical details are clear. They conduct extensive experiments to demonstrate the efficacy and efficiency of the designed solution.

---

### Official Review · Reviewer_ouFd · 2022-11-30

**Confidence:** 4
**Correctness:** 4
**Technical Novelty And Significance:** 3
**Empirical Novelty And Significance:** 4
**Recommendation:** 8

**Clarity, Quality, Novelty And Reproducibility:**

The work is clearly written and easy to follow. The main technical contributions are novel.

**Strength And Weaknesses:**

Clearly the meta knowledge sharing in place of the classic model weight aggregation is the most important and strong contribution of this work. In so heterogeneous scenarios in federated systems, model weight aggregation is really the pain point and potentially very bad in terms of communication efficiency. However, this work pointed out an alternative with clear empirical evidences that support the superiority of  it.

The only weakness I found in this work is the lack of discussion about the influence of the proposed condensation mechanism on the privacy preserving property of FL. This is crucial because the local meta knowledge will finally be sent to the cloud. While the authors didn't include discussions on privacy in the draft, I highly encourage them to at least add some references to existing analysis work about dataset condensation, which is not new technically in non-FL scenarios.

**Summary Of The Paper:**

This paper proposed a new learning scheme in Federated Learning that communicates knowledge between clients in the form of meta knowledge generated by dataset condensation. In this case, the global model aggregation in the cloud is replaced with a "meta training" process in the cloud using the combined meta knowledge from all participating clients in each communication round. The authors also proposed several modules that co-work with the main meta knowledge scheme, such as pseudo knowledge in the cloud using a generating model and dynamic weights.

The idea is clearly novel and the empirical effect of reducing communication rounds needed is impressive. The authors also show that other designed modules all contribute to the superiority of the overall framework.

**Summary Of The Review:**

This is an outstanding work with novel technical contributions and impressive empirical performances.

=====================================

Note for the late submission of this review. I am very sorry that I submitted my review this late. I think I wrote the review locally on my computer but forgot to post it to OpenReview.

My original evaluation on this work without referring to other reviews was very positive, with only  some minor issues on the math formulation and some statements, such as:

1. In equations (4,5,7) of the original draft and the equations (3,5,6,8) of the updated draft as of 11/30/2022, the $\nabla$ operator misses subscripts that indicate with respect to which the derivatives are taken. The subscripts are needed because the derivatives are taken w.r.t. different inputs to the loss function $\mathcal{L}^c$.
2. In equation (6) of the original draft, the trailing parenthesis is missing. It is still missing in the updated draft.
3. The $\mathcal{L}^c$ notation is abused in the equation (7) of the original draft. The authors should mention that the notation comes with a little abuse.
4. The authors said above equation (7) in the original draft that "Apparently, the weight of each sample is inversely proportional to its prediction loss", which is contrary to the formulation. It seems to have been fixed in the updated draft.
5. There has been some unclarity of the bilevel optimization formula, but it has been addressed in the discussions between the authors and other reviewers.

I encourage the authors to address the above points, while they have not influenced my overall evaluation of this work so I hope it won't make the rebuttal process messy.

---

### Public Comment · ~Tiandi_Ye1 · 2023-02-20
**Privacy Leakage and MISSING OF A LOT OF RELATED WORK**

1. This work does not consider and discuss the problem of privacy leakage caused by dataset condensation.
2. The current version of this paper misses A LOT OF related works, which highly overlap with the proposed FL+dataset distillation paradigm[1,2,3,4,5].

[1] Goetz, Jack, and Ambuj Tewari. "Federated learning via synthetic data." arXiv preprint arXiv:2008.04489 (2020).

[2 ]Hu, Shengyuan, et al. "Fedsynth: Gradient compression via synthetic data in federated learning." arXiv preprint arXiv:2204.01273 (2022).

[3] Behera, Monik Raj, et al. "Fedsyn: Synthetic data generation using federated learning." arXiv preprint arXiv:2203.05931 (2022).

[4] Song, Rui, et al. "Federated learning via decentralized dataset distillation in resource-constrained edge environments." arXiv preprint arXiv:2208.11311 (2022).

[5] Kim, Seong-Woong, and Dong-Wan Choi. "Stable Federated Learning with Dataset Condensation." J. Comput. Sci. Eng. 16.1 (2022): 52-62.

---

> ### Author Response · Authors · 2023-02-20
> **The reply**
>
> Thank you for your comments.
>
> About your first comment, we discussed the issue of privacy leakage in our initial supplementary materials ("The possibility of restoring original data from meta-knowledge"), and provided a related response to reviewer ouFd (e.g., we conducted discussions about Privacy for free: How does dataset condensation help privacy? ICML 2022).
>
> Regarding your second comment, we compared our work with the relevant studies, including "Distilled one-shot federated learning" (arXiv 2020), Kim et al. [5], and FedMix (ICLR 2021), which can be found in our initial submission and responses to the AC and Reviewer.
>
> Based on the suggestions and comments from AC and Reviewers, in the camera ready version, we have included more relevant works (e.g., FedDM, NeurIPS workshop 2022; Fedsynth, NeurIPS workshop 2022; Fedsyn, arXiv 2022) and briefly discussed the differences. Please refer to Related Work section. Correspondingly, we made adjustments to our contributions based on the suggestions from the AC and Reviewers.

---

### Public Comment · ~Chen_Ruoxi1 · 2023-03-01
**Confusion about communication cost calculation**

In Appendix A, the communication cost calculation is introduced as follows：
First of all, with MetaKnowledge is uploaded and downloaded once and the model is downloaded once, the calculation of communication cost should take in account of the number of active nodes, the communication round and the size of MetaKnowledge.
Take the MNIST dataset as an example.
（1）the uploading cost is 4Bytes x 28 x 28 x 20 x 10 x 10 x 10(#Image Size x #MetaKnowledge per class x # class x # active node x # round, and you missed the active node!), which is 62720KB, which is larger what it is in manuscript. So, it may be a mistake!
（2）the downloading cost is 105K x 10 x 10 + 62720KB, which is 73,220KB.
Therefore, the communication cost is 135.94MB. And for CIFAR100 is 665.86MB! Compared to 420MB for FedAvg, the cost brought by MetaKnowledge transferring is huge!!

Second, the comparison is a little unfair. Since transferring data enables fast convergence of the global model, the cost of communication is greatly reduced. Can analyze whether the knowledge is redundant？

---

> ### Author Response · Authors · 2023-03-01
> **The reply**
>
> Thank you for reaching out with your inquiries. We are pleased to have the opportunity to provide further clarification.
>
> In regards to the first question, the number of active clients was already considered in our calculation. As a result, the introduction of an additional factor of "x10" was not deemed necessary. Our analysis is based on the premise that each client can only distill the meta knowledge for the class(es) present on it.
>
> Our data sampling and distribution was carried out through a Dirichlet distribution with small alpha values, leading to a limited number of classes being distributed on each client. In such instances, where only a small portion of classes are present on a client, it can only distill meta knowledge for those classes (e.g., when alpha is 0.1, it becomes impossible for each client to distill meta knowledge for all classes). This explains the presence of only two instances of "x10" in the calculation, with one denoting the communication round number and the other denoting the class number.
>
> We note that in scenarios where the alpha value is larger, for example, 10, or the data is evenly distributed among all clients, each client will have the capacity to distill meta knowledge for all classes. In such cases, it would be necessary to introduce a third "x10" in the calculation, making the communication cost larger. However, as outlined in Introduction, the focus of our work is to enhance the efficiency of Federated Learning when communication budgets are constrained, i.e., in the context of 10 communication rounds, instead of reducing the bandwidth of data transmission.
>
> With regards to the second question, we conducted an experiment to investigate the potential redundancy in meta knowledge. For further details, please refer to Appendix B, "Why Increasing the Meta Knowledge Size Does Not Necessarily Improve Final Performance?" Our experiment on the MNIST dataset aimed to demonstrate the variations in information content of meta knowledge with respect to different sizes. The results of the experiment showed that the amount of information contained in the meta knowledge does not exhibit significant changes with respect to its size.
>
> We hope that this additional information has been helpful in addressing your inquiries.

---

### Decision · Program_Chairs · 2023-01-20

**Decision:**

Accept: poster

**Justification For Why Not Higher Score:**

- Novelty is relatively weak, since similar approach has been proposed in previous works such as [Yoon et al. 21] and [Kim et al. 22].
- The privacy concern is not fully resolved.

**Justification For Why Not Lower Score:**

The proposed method obtains impressive performance over compared baselines.


**Metareview: Summary, Strengths And Weaknesses:**

This paper proposes a federated learning framework which communicates condensed datasets obtained using meta-learning, which are then used as pseudo-training examples to train a global model at the server, along with the synthetic samples generated by a generator that is trained with the pseudo-training examples. The method is validated on multiple benchmark datasets with multiple federated learning algorithms, and is shown to obtain significant improvements over the base models.

The reviewers in general considered the idea of communicating meta-knowledge instead of model parameters as novel, the proposed dataset condensation and dynamic weight assignment for obtaining meta-knowledge from each client as reasonable, the experimental validation including the ablation studies as extensive, and the paper well-written.

However, the reviewers also had a common concern on the potential data leakage from the condensed datasets that are transmitted to the server. In response, the authors provided some experimental results which show that it is very difficult to restore the original data from the meta-knowledge, and referred to the theoretical analysis in [Dong et al. 22], which dealt away with some of their concerns. As this resolved the critical concerns from the reviewers, all reviewers recommended to accept the paper after the discussion period.

There was also an additional concern on my side on the lack of discussion and experimental comparison against a high related work, FedMix [Yoon et al. 21], which leverages averaged data from each client as well as augmented samples generated with mixup regularization, and [Kim et al. 22] which proposes a federated learning framework based on dataset condensation. The authors provided some discussions and experimental comparisons against them in the interactive discussion period, as well as some experimental results. I strongly suggest them to include the discussions as well as full experimental comparisons against them in the final revision, and adjust the claimed contributions of the proposed work accordingly. Also, since data leakage could be critical for a federated learning scenario, there should be a more extensive analysis of the potential threats to the data privacy introduced by the proposed method.

[Yoon et al. 21] FedMix: Approximation of Mixup Under Mean Augmented Federated Learning, ICLR 2021
[Kim et al. 22] Stable Federated Learning with Dataset Condensation, JCSE 2022

**Note From Pc:**

if the above contains the word "oral" or "spotlight" please see: "oral" presentation means -> notable-top-5% and "spotlight" means -> notable-top-25%. As stated in our emails, we are disassociating presentation type from AC recommendations